# Toxic Threats from the Fern *Pteridium aquilinum*: A Multidisciplinary Case Study in Northern Spain

**DOI:** 10.3390/ijms26157157

**Published:** 2025-07-24

**Authors:** L. María Sierra, Isabel Feito, Mª Lucía Rodríguez, Ana Velázquez, Alejandra Cué, Jaime San-Juan-Guardado, Marta Martín, Darío López, Alexis E. Peña, Elena Canga, Guillermo Ramos, Juan Majada, José Manuel Alvarez, Helena Fernández

**Affiliations:** 1Department of Functional Biology (Genetics Area), University of Oviedo, C/Julián Clavería s/n, 33006 Oviedo, Spain; lmsierra@uniovi.es (L.M.S.); uo283552@uniovi.es (A.V.); alecue8@gmail.com (A.C.); martamj25@gmail.com (M.M.); dariolpzlopez@gmail.com (D.L.); 2University Institute of Oncology of Asturias (IUOPA), University of Oviedo, 33006 Oviedo, Spain; 3Health Research Institute of Asturias (ISPA), Avda. HUCA s/n, 33011 Oviedo, Spain; 4The Regional Agri-Food Research and Development Service (SERIDA), 33630 Asturias, Spain; isabelfeito.diaz@asturias.org (I.F.); marialucia.rodriguezperez@asturias.org (M.L.R.); 5Area of Plant Physiology, Department BOS, c/ Catedrático R. Uría s/n, 33071 Oviedo, Spainalvarezmanuel@uniovi.es (J.M.A.); 6Forestry and Wood Technology Centre (CETEMAS), 33936 Asturias, Spain; ecanga@cetemas.es (E.C.); grgonzalez@cetemas.es (G.R.); jmajada@cetemas.es (J.M.)

**Keywords:** bracken, *Pteridium aquilinum*, pterosins A and B, in vivo genotoxicity, SMART assay of *Drosophila melanogaster*, UHPLC-MS/MS, UAV drones

## Abstract

*Pteridium aquilinum* (bracken fern) poses a global threat to biodiversity and to the health of both animals and humans due to its toxic metabolites and aggressive ecological expansion. In northern Spain, particularly in regions of intensive livestock farming, these risks may be exacerbated, calling for urgent assessment and monitoring strategies. In this study, we implemented a multidisciplinary approach to evaluate the toxicological and ecological relevance of *P. aquilinum* through four key actions: (a) quantification of pterosins A and B in young fronds (croziers) using ultra-high-performance liquid chromatography coupled with tandem mass spectrometry (UHPLC-MS/MS); (b) analysis of in vivo genotoxicity of aqueous extracts using *Drosophila melanogaster* as a model organism; (c) a large-scale survey of local livestock farmers to assess awareness and perceived impact of bracken; and (d) the development and field application of a drone-based mapping tool to assess the spatial distribution of the species at the regional level. Our results confirm the consistent presence of pterosins A and B in croziers, with concentrations ranging from 0.17 to 2.20 mg/g dry weight for PtrB and 13.39 to 257 µg/g for PtrA. Both metabolite concentrations and genotoxicity levels were found to correlate with latitude and, importantly, with each other. All tested samples exhibited genotoxic activity, with notable differences among them. The farmer survey (n = 212) revealed that only 50% of respondents were aware of the toxic risks posed by bracken, indicating a need for targeted outreach. The drone-assisted mapping approach proved to be a promising tool for identifying bracken-dominated areas and provides a scalable foundation for future ecological monitoring and land management strategies. Altogether, our findings emphasize that *P. aquilinum* is not merely a local concern but a globally relevant toxic species whose monitoring and control demand coordinated scientific and policy-based efforts.

## 1. Introduction

The *Pteridium aquilinum* fern is now recognized as a complex taxonomic group that includes at least 17 subspecies and varieties [1], with *P. aquilinum* var. *aquilinum* being one of the most widespread, found on every continent except Antarctica [2,3]. All plants within this genus are commonly referred to as bracken, a term that will be used henceforth for brevity. The prevalence of bracken in the world is very important, occupying forest ecosystems, grasslands, and shrublands [3]. The overall geographical distribution and local abundance of bracken appear to be increasing in various regions around the world [4], driven in part by human land-use practices but also by its natural aggressiveness towards other species, including plants, insects, and herbivores [5,6,7,8,9,10]. Climate change facilitates its spread, particularly in Northern Europe and mountainous areas, due to rising temperatures, longer growing seasons, increased humidity, and greater sunlight exposure [4,10]. This plant is seen as a potential threat to biodiversity because it can replace important habitats, such as species-rich grasslands [3,11], and it disrupts the regeneration of hardwood forests [11,12,13], hindering the establishment and growth of tree seedlings [12]. These effects are mainly due to the vast array of chemicals produced by this plant [14,15,16,17,18].

The most studied compounds include cyanogenic glycosides, iludane glycosides (ptaquiloside (PTA), caudatoside (CAU), and ptesculentoside (PTE)), several pterosins, and selligueain A, a germination inhibitor [19,20,21]. Iludane glycosides have gained attention because they are the source of the bracken carcinogenicity [14,15,19,21,22,23,24,25,26,27,28,29,30,31]. Bracken is the only known vascular plant that can naturally induce cancer in animals and humans [18,32,33,34,35], and it is classified as “possibly carcinogenic to humans” (Group 2B) by the International Agency for Research on Cancer [36].

PTA is the most well-studied chemical of this group, although CAU and PTE are expected to have the same characteristics and properties [28,30,31]. PTA was discovered in 1983 [37,38] and identified as a mutagenic chemical and as “the carcinogenic metabolite of Pteridium” [38,39,40]. PTA is an indirect mutagen, and its reactivity is enzyme-independent but pH-dependent [28,41]. In base conditions, the D-glucose moiety is lost from the PTA, and an unstable and highly reactive conjugated dienone is formed [24,28,34]. This dienone alkylates DNA, at nitrogen and oxygen atoms, and it is accepted as the activated metabolite of PTA [22,25,34,39,41,42]. In acid conditions, the D-glucose moiety is also lost and a pterosin is formed (specifically pterosin B, PtrB), which is not a genotoxic metabolite [16,17,25,28,43]. This stable PtrB can also be formed from the dienone metabolite [25,28,30]. The distribution of these toxins varies between the fronds and rhizome of the plants, with the highest concentrations found in the fronds, especially during the early growth stages (croziers) and at the tips of the pinnae [20,31,44,45,46]. Additionally, spores also contain at least PTA [31,46], and they also induce DNA adducts [22,25] and DNA strand breaks [47].

Both the plant and PTA can induce cancer in cattle; specifically, urinary bladder cancer in a syndrome known as bovine enzootic haematuria (BEH) [14,15,16,41].

In humans, epidemiological studies have revealed elevated risk of tumours in the digestive system of persons who consume bracken, a finding observed in various regions around the globe [24,32,34,41,48,49]. In areas where bracken is not part of the diet, potential exposure sources include milk and meat from cattle that have ingested the bracken [16,34,50] and inhalation of spores or dust [47,51]. In addition, since bracken metabolites PTA, CAU, and PTE are highly water-soluble, they can leach from the fronds into the soil and, subsequently, permeate water sources and troughs, including groundwater and drinking water [20,34,52,53,54,55].

In Spain, the Autonomous Region of Asturias, together with Galicia, Cantabria, and the Basque Country, throughout the Cantabrian north coast, and Extremadura in the south-west, constitutes a heterogeneous phytogeographic arc, characterized by the exploitation of cattle, sheep, and horses, and in which bracken stands out for its considerable abundance and coverage of the landscape. However, despite the quite important risk represented by this plant, no comprehensive studies have been performed to determine the actual hazard that Pteridium plants present.

Because of this lack of information, this work aimed to determine the actual threat posed by *Pteridium aquilinum* in Asturias. To achieve this objective, four detailed tasks were outlined. The first was the determination of the iludane glycoside pterosins, as stable chemicals, in samples of croziers taken from plants located in diverse environments all over Asturias, especially in areas with significant livestock activity and/or regions of notable wildlife interest (like national and natural parks). Mass spectrometry was the analytical methodology chosen to perform this task, as described previously [56]. The second task was to study the in vivo genotoxicity of aqueous extracts obtained from these plants, using *Drosophila melanogaster* as a model organism, to avoid the use of mammals [57], and the SMART assay, which detects induction of mutation and recombination in somatic cells of *Drosophila* larvae [58,59,60]. In this SMART assay, wild-type eyes may present *white* mutant spots as a consequence of DNA damage: in females, DNA damage can give rise to mutant spots mainly through its fixation as somatic mutation or mitotic recombination events, whereas in males, DNA damage originates mutant spots almost exclusively through its fixation as somatic mutation [59,61,62]. The genotoxic activities detected in this assay were compared to the levels of the analysed metabolites. The third task was to engage farmers through an in-depth survey to gather first-hand accounts and experiences that can further inform the understanding of the issue. Finally, the last task was to design and develop a methodology to map this species at a regional level, using drones or unmanned aerial vehicles (UAVs), and to check it, in a proof-of-concept study, to analyse the multispectral data provided by the UAV flights.

The results demonstrated that, although there were differences among plants both at the levels of metabolites and at the respective genotoxic activities, pterosins PtrB and PtrA were detected in all of them, and all the samples were genotoxic in vivo in *Drosophila larvae*. Furthermore, the levels of genotoxic activity strongly correlated with the levels of PtrB and PtrA. The conducted survey revealed that not all the farmers were aware of the danger posed by this plant. Lastly, a methodology was developed to map the spread of this plant, using UAV flights.

## 2. Results

### 2.1. Quantification of Toxic Metabolites in Croziers Samples

The results of the implemented analytical methodology used to determine levels of *P. aquilinum* metabolites, in terms of sensitivity and accuracy, as well as the correlation coefficients of the calibration curves performed for each of the analysed pterosins, are presented in Table 1. As indicated, only the pterosins A and B were determined because they are very stable chemicals, and reliable standards were available.

As observed, values of LODs and LOQs were low enough to make a reliable determination of metabolites, and high correlation coefficients (R^2^) of the calibration curves were obtained for the respective concentration ranges.

With this methodology, the levels of these three metabolites were determined in extracts from plant samples, specifically croziers, and the corresponding results are presented in Figure 1. These results show that these two metabolites were found in all the samples.

In addition, these results show clear differences among the analysed samples for each metabolite. As observed, levels of PtrA were between 6 and 260 µg/g dry weight (DW), and levels of PtrB were around 10 times higher than those of PtrA (between 55 and 2300 µg/g DW). In addition, levels of PtrB and PtrA follow a quite similar distribution among samples, despite the different scale (correlation *R =* 0.94, *p* < 0.000).

Statistical analysis to check differences among samples, for each metabolite, was carried out with one-way ANOVA and a posteriori SNK analyses. The results indicated statistically significant differences for all the metabolites (*p* < 0.000). However, for each of them, there were some homogeneous subgroups revealed by the SNK analyses, and again, similarities were found between PtrB and PtrA (Appendix A).

To provide detailed insights into these findings, correlations and regression analyses were performed between mean levels of metabolites and the geographic parameter latitude of all the samples. As shown in Figure 2, positive statistically significant correlations (12 d.f.) and regression slopes were found for PtrB and PtrA.

A negative, almost statistically significant, correlation was also found between the levels of PtrB and the altitude (*R =* −0.53, *p* = 0.051). No relationships were found with longitude coordinates.

### 2.2. Genotoxicity Analysis of Aqueous Plant Extracts

The SMART assay of *D. melanogaster* was used to check the possible genotoxicity of aqueous extracts of Pteridium plants (Figure 3).

The results of the hatched flies per bottle, used as a semiquantitative measurement of toxicity, presented in Figure 3A, showed the lack of differences between the extract concentrations and the corresponding negative controls, for each sample, with two exceptions: the lowest analysed concentration of the PELH sample and the middle concentration of the PON sample.

The frequencies of mosaic eyes in females are presented in Figure 3B. The extracts from all the samples, at concentrations of 10 and 25 mg/mL, induced frequencies of mosaic eyes significantly higher than those of the corresponding negative controls. The extracts from PRI, LaM, MU11, MU13, MU2, and MU15 samples, at 1 mg/mL concentration, did not increase the frequency of mosaic eyes over their respective negative controls.

In males, the results of the frequency of mosaic eyes are presented in Figure 3C. All the extracts at 25 mg/mL concentration induced frequencies of mosaic eyes significantly higher than the corresponding negative controls, as did all the samples but DEG at 10 mg/mL concentration (*p =* 0.0825 with correction for continuity factor), and all but PELH, PON, LaM, DEG, and all the MU samples at 1 mg/mL concentration.

The comparison between females and males revealed higher frequencies of mosaic eyes in females than in males for all the samples. However, in both sexes, clear differences were detected among the samples. The induced frequency of mosaic eyes in males represents the induction of somatic mutations, whereas the difference between the induced frequencies of mosaic eyes in females and males represents the induction of mitotic recombination [59,61,62]. The comparison between induction of mutation and recombination, presented in Appendix A, with large differences among samples, showed that although in general the frequencies of induced mutations were larger than those of recombination, especially with the higher extract concentrations, some samples induced more recombination events than somatic mutations, like MU15, LaM, and MU13.

These data on genotoxic activities were analysed together with the geographic parameters. The results of the correlations and regression analyses with latitude are presented in Figure 4.

Positive statistically significant correlations and regression slopes were found for the mosaic eye frequencies induced by 1 and 10 mg/mL extract concentrations in males and by 10 mg/mL extract concentration in females.

No relationships were detected between the induced frequencies of mosaic eyes and the longitude.

The analysis of the relationships with altitude revealed a negative significative relationship with the mosaic eye frequency induced by 10 mg/mL extract concentration in females, but only when the PRI sample was removed from the analysis (*R =* −0.55, with 13 d.f., *p* = 0.034).

Additionally, the pH and viscosity of the aqueous extracts (Appendix A) were compared to altitude and latitude. The results showed negative and statistically significant correlations (with 14 d.f.) between viscosity and altitude (*R =* −0.54, *p* < 0.05) and between pH and latitude (*R =* −0.62, *p* < 0.05).

### 2.3. Relationships Between Pterosin Levels and Genotoxic Activities

The levels of PtrA and PtrB were checked against the induced frequencies of mosaic eyes, in both sexes, for the studied samples, again using correlation and regression analyses. The results of the regression analyses for PtrB are presented in Figure 5.

Regression and correlation analyses, with 12 d.f., showed R values of 0.74 (*p =* 0.002), 0.87 (*p =* 0.000), and 0.70 (*p =* 0.005) for 1, 10, and 25 mg/mL extract concentrations, respectively, in females (Figure 5A) and 0.52 (*p =* 0.054), 0.65 (*p =* 0.012), and 0.65 (*p =* 0.012) in males (Figure 5B), indicating positive relationships between the analysed parameters, all of them statistically significant, except that corresponding to 1 mg/mL in males, which is close to significance.

Results of regression and correlation analyses for PtrA, with 12 d.f., showed R values of 0.77 (*p =* 0.001), 0.88 (*p =* 0.000), and 0.70 (*p =* 0.005) for 1, 10, and 25 mg/mL extract concentrations, respectively, in females (Figure 6A) and 0.67 (*p =* 0.009), 0.78 (*p =* 0.001), and 0.74 (*p =* 0.003) for these same concentrations in males (Figure 6B), indicating positive and statistically significant relationships between the levels of PtrA and the induced frequencies of mosaic eyes.

Both the regression and the correlation analyses revealed stronger relationships of the induced frequencies of mosaic eyes with PtrA than with PtrB.

To check that these very strong relationships were not due to an as yet undetected genotoxic activity of pterosins PtrA and PtrB, SMART assays were performed with 0.02 mg/mL of PtrB and 0.002 mg/mL of PtrA, concentrations close to the highest detected ones in the 10 mg/mL concentration of the aqueous extracts. The results of these experiments with frequencies of mosaic eyes of 7.96 and 10.94 in females, and 3.83 and 4.38 in males, for PtrA and PtrB, respectively, demonstrated their lack of genotoxic activity when compared to the negative control frequencies of 10.61 and 4.06 in females and males, respectively (with more than 300 scored eyes per sex).

To obtain more information about the relationships between metabolites and genotoxicity, the levels of PtrA and PtrB were checked against the induced frequencies of somatic mutations and mitotic recombination events. Only relationships between this parameter and the levels of PtrB were detected (regression and correlation analyses, with 12 d.f., *R =* 0.60, *p =* 0.023, and R = 0.61, *p =* 0.02, for 1 and 10 mg/mL extract concentrations, respectively) (Appendix A).

Concerning relationships with induced mutations, since information about the induction of somatic mutations is obtained from the male data, it is clear that the levels of both pterosins were strongly related to the induction of somatic mutations (Figure 5B and Figure 6B).

To check whether the relationships between PtrB and PtrA levels and genotoxic activities could be improved in such a way that the pterosin levels might be used to predict genotoxic activities, multivariate regression analyses were performed combining PtrB and PtrA levels. The results showed that only for the mosaic eye frequencies induced by the 1 and 10 mg/mL concentrations, in males, did the relationships increase—to *R =* 0.725 (*p =* 0.016) and *R =* 0.824 (*p =* 0.002), respectively—with the combination of these pterosins. Moreover, when latitude, altitude, extract pH, and viscosity parameters were included in the analyses, the relationships increased to *R =* 0.901 (*p =* 0.026) and 0.919 (*p =* 0.014) for 1 and 10 mg/mL extract concentrations in females. For the 25 mg/mL concentration, the relationship increased to 0.865 (*p =* 0.026) when viscosity was excluded from the analysis. In the case of the induced frequencies of mosaic eyes in males, the relationship increased for the 10 mg/mL concentration to *R =* 0.94 (*p =* 0.006). For the 25 mg/mL concentration, the addition of latitude, altitude, and extract viscosity parameters, without the extract pH, also increased to *R =* 0.849 (*p =* 0.037). For the 1 mg/mL extract concentration, the addition of these other parameters did not increase the relationship.

### 2.4. Results of Farmer Surveys

The results of the surveys that covered all the parts of Asturias where plant samples were collected are summarized in Appendix A. From the collected responses, the first point of interest was that beef cattle farming is very important in Asturias. While these animals were the primary livestock, other species such as sheep, goats, and horses were also present in the surveyed farms, in lower numbers, although a significant presence of sheep and goats was recorded in the area of PE. In the natural park LaM, only one farmer, out of 24, raised goats (more than 100). The number of cows owned per farmer remains consistent, ranging between 43 and 90, and the Kruskal–Wallis test results revealed no significant differences among the tested zones (H(8) = 11.82, *p =* 0.159).

The survey results revealed that a large majority of farmers were aware of the potential damage that ferns pose to their cattle, and some of them even reported witnessing cattle consuming the plant. However, significant differences were detected between the sampled areas (Chi square X^2^ = 41.404; *p* < 0.05). Notably, farmers from DEG, as well as those from NOV and TIN, exhibited a concerning lack of knowledge regarding the risks associated with bracken.

Concerning animal illness and deaths, the survey revealed that the symptom of blood in urine is alarmingly common across all sampled localities, with around half the farmers having observed it at some point and without significant differences among the sampling sites (X^2^ = 11.671, *p* = 0.1665).

Concerning the percentages of farmers reporting livestock deaths attributable to toxicity or poisoning, the highest values were reported in PE and PON, with 48.27% and 47.82% of the surveyed farmers, respectively, whereas in DEG, NOV, and PRI, no deaths were linked to bracken poisoning. However, the statistical analysis revealed that these differences did not reach statistical significance (X^2^ = 12.298, *p =* 0.138). No significant differences among locations were detected in the number of dead cows according to the Kruskal–Wallis test (H(8) = 11.557, *p =* 0.1721). These records reflect not only isolated incidents but, in many cases, recurrent issues faced year after year. Importantly, 100% of reported deaths in PE, PON, SOM, and TIN were certified by veterinarians to have been caused by bracken consumption, particularly during the late summer and autumn months (Appendix A). The survey also included questions to farmers about their practices concerning silage. The majority of them reported exercising caution to prevent bracken remnants from being included in their forage packages. Furthermore, the farmers expressed their deep concern regarding the encroachment of this plant on their grazing lands. Many acknowledged the urgent need to control its growing expansion and mentioned the use of herbicides (those allowed) and the manual cutting of the plant fronds as their main strategies for management.

### 2.5. Implementing a Tool to Map Bracken

As indicated in the Introduction, the last partial objective of this work was to design a methodology to map bracken plants at the regional level using UAVs. For that, UAVs were flown at different times throughout the year. With the data collected in the first of those flights, a significant difference was found among the spectral reflectance of different vegetation types (Appendix A).

Afterwards, other UAVs were flown to check the methodology with training and tests. The results showed that a Kappa index of 0.91 and an overall accuracy (OA) of 0.94% were obtained, which indicated promising features of the methodology. With this achievement, test flights were carried out. In this case, the obtained results showed a Kappa index of 0.69 and an overall accuracy (OA) of 0.80%. Appendix A show the confusion matrix in the case of training of the random forest classifier and the test flights, respectively.

With this methodology of acceptable accuracy, a vegetation map was produced, which is presented in Figure 7.

As observed, the bracken is identified in the map much better and more accurately than in the original image. Figure 7 also shows the large expansion of bracken in the monitored area.

## 3. Discussion

This work aimed to conduct the first comprehensive assessment of bracken in Asturias, as one of the Spanish regions with a large presence of this plant. To achieve this aim, several sampling locations were chosen, from the coast to the high mountains, to create a data set that would provide information to help people decide on effective management strategies.

The findings of this work indicate that at least pterosins PtrB and PtrA are consistently present in the croziers collected from all sampled locations, revealing therefore the presence of their respective iludane glycosides PTA and CAU, theoretically in equivalent levels. It has been described that PTA is distributed throughout the entire plant, with varying concentrations in different parts: from lowest to highest, in spores, roots, rhizomes, and fronds [46,63,64,65]. Moreover, the croziers represent the phenological stage where PTA concentration is quite high, although it decreases throughout the growing season [66,67].

PTA has received more attention in research than CAU or PTE, as data on these two compounds are more limited [31]. Moreover, a higher prevalence of PTA over CAU and PTE was described in plants from several regions of the world [30,31,66].

The analysis of pterosins in this work demonstrated a noteworthy correlation between the levels of PtrA and PtrB in the analysed plants, suggesting that PTA and CAU are present in high levels in these plants and that they follow a similar pH-dependent transformation. To understand the variations observed among these sampling locations, it is essential to consider the multitude of factors that can influence this type of transformation, which may arise from the characteristics of the plant itself or from environmental conditions [31,68].

Although the levels of the two pterosins showed statistically significant relationships with latitude, and those of PtrB also with altitude, the many different homogeneous groups detected for the levels of PtrA and PtrB suggest that other factors might also be involved in the synthesis of iludane glycosides. However, the different soils, with different pHs, present in the eastern and western parts of Asturias, related to the different types of rocks (limestone and carbonate rocks in general in the east and sandstone, quartzite, granite, and slate in the west [69])**,** do not seem to be important, because no relationships were found between metabolite levels and longitude. Some environmental factors were described to be important in the production of iludane glycosides, mainly nutrient viability, particularly phosphorus, organic matter content, soil pH, and altitude [41,70,71,72]. Recent studies quantifying PTA, CAU, and PTE over a broad geographical range in Northern Europe have revealed a complex interplay of climatic and genetic factors affecting secondary metabolite production in bracken [31]. However, to our knowledge, this is the first time that strong relationships between metabolite levels and latitude, in a rather small area, were detected.

The risk of exposure to bracken highlighted by the levels of the pterosins was confirmed when the genotoxic activities of aqueous extracts of the samples were demonstrated in in vivo experiments with *D. melanogaster*. These genotoxic activities were different among the samples, and also regarding their potential to induce mutation or recombination events (Appendix A); moreover, they were induced without toxicity, because the two toxic treatments, with low extract concentrations, were not considered to be biologically relevant. These results are in concordance with previous reports of the genotoxic activity of bracken extracts determined with different genotoxicity assays. Starting with the induction of DNA adducts both in different animals in vivo [22,25,41] and in vitro [25,41], there is information about the induction of DNA strand breaks, determined with the comet assay in vitro [47,73,74] or through γ-H2AX [75]. Furthermore, bracken extracts were reported to induce chromosomal aberrations in vivo [28,35,76,77,78]. Equivalent results were obtained with PTA treatments, which induced DNA adducts in vivo [79], DNA strand breaks both in the comet assay in vitro in mononuclear blood cells [28] and in γ-H2AX in cultured human cells [75], and chromosomal aberrations in vitro [40]. In addition, PTA induced gene mutations in bacteria [41,80,81].

Concerning gene mutations in mammalian cells, whereas some evidence indicated that PTA induces mutations at the H-ras gene in rats in vivo [82,83], no mutations in this gene were detected in immunohistochemical analysis of urinary bladder lesions from slaughtered cattle [84], nor on the TP53 gene in cattle tumours [85]. With this evidence, [30] suggested that no real evidence exists that PTA or bracken extracts induce mutations in ras or TP53 genes. However, a mutation signature for PTA has very recently been described [86].

The genotoxic activities of the analysed samples, like the metabolite levels, presented a clear relationship with geographic locations, specifically with latitude. Most importantly, they presented strong relationships with the levels of PtrB and PtrA. Since these pterosins are not genotoxic, as shown in this work and as reported in the literature [16,17,25,30,41,43], the detected relationships should be related to PTA and CAU, and their respective dienones, which generate these pterosins [16,30,39,41,68,87].

This is the first time that such relationships between genotoxicity and specific bracken metabolites have been described, and they emphasize the relevance not only of PTA but also of CAU in the genotoxicity and therefore in the carcinogenicity of Pteridium plants in Asturias. The relationship of PtrA with mutation induction but not with recombination induction points to an important role of CAU in the induction of somatic mutations in *D. larvae*, whereas PTA seems to be important in the induction of both types of events. An increased role of CAU in *P. aquilinum* plants in Europe has recently been suggested [31].

Over the past two years, a comprehensive study involving 212 surveys targeting farmers in the chosen locations was conducted. To enhance its outreach, active participation in livestock fairs, held across various municipalities in Asturias, were carried out from spring to winter. The primary goal of this part of the work was to engage with farmers who manage extensive livestock operations, focused on meat cattle that graze in open fields. This approach increased the likelihood of obtaining information related to bracken consumption. Through the interviews, it has become clear that many farmers are aware of the potentially harmful effects of bracken on livestock, especially cows, although they lack detailed knowledge about it. Nevertheless, farmers from the NOV, PRI, TIN, and DEG locations seemed to be less familiar with the danger of bracken.

When asked about symptoms related to bracken poisoning, approximately 25% of the farmers reported having observed blood in the urine of cows at some point. However, many of them attributed these symptoms to tick bites. This association is plausible, as bracken habitats are often linked with tick presence [31,88], which might transmit important tick-borne diseases [89,90]. This trend appears to be particularly pronounced at the NOV, PRI, and DEG locations. Although no other symptoms related to bracken poisoning were detected, about half the surveyed farmers acknowledged the loss of some cows to bracken consumption, certified by veterinarians, with numbers per farm between 1 and 15 in 10 years; the less affected areas were NOV and SOM, and the most affected ones were PE and PON, where veterinarians have attributed the recorded deaths to bracken consumption, primarily occurring in late summer and autumn.

There is a growing consensus among farmers in the most impacted areas that harsh conditions, such as reduced pasture availability, hot summers, or increased livestock pressure, raise the risk of livestock death due to bracken consumption. One farmer, from the LaM natural park, lost 19 cows in two months (October and November) more than 10 years ago. Addressing this problem is crucial, since farmers are increasingly referring to the problems associated with bracken consumption as “bracken disease”.

Moreover, farmers expressed significant concern about the loss of pasture areas due to the aggressive expansion of bracken. Looking ahead, especially in the context of climate change, it is crucial to explore effective means of controlling its spread. To date, farmers in Asturias have been cutting back the aerial parts of the plant and using systemic herbicides, and some of them have carried out the mechanical removal of the rhizomes or the burning of the plants. With respect to the risk of bracken ingestion from silage, most farmers tried to prevent its inclusion in the packages.

Finally, concerning the implemented methodology to accurately map bracken in vegetation maps of Asturias, the higher reflectance values of bracken compared to those of grass, in the red part of the spectrum, because of its richness in dead material, have allowed the detection of a greater difference in NIR band [91]. Similar results were obtained for the precision statistics (OA and Kappa index) in other studies [9,92]. In the case of bracken, the producer accuracy was 0.94% in training, decreasing to 0.80% in tests. These results demonstrate that an RF model for vegetation type classification has been developed and included in an automatic mapping tool. The open-source tool generates a land cover map using UAV multispectral data obtained two times in a year and a free LiDAR point cloud. The results show the potential of multispectral data in mapping this invasive species, but it will be necessary to increase field data and to include more flights throughout the year to improve the RF model and its scaling for satellite data.

Summarizing, the work presented here demonstrated the clear and direct risks posed by bracken plants, because their metabolite contents are linked to their genotoxic activities. The complexity of these relationships underscores the importance of conducting further studies to clarify them and to enhance the understanding of this subject. This bracken risk is a reality to cattle health, as revealed by the surveys carried out among farmers.

## 4. Materials and Methods

### 4.1. Sampling Sites and Pteridium Plant Collection

This study has been conducted across ten locations in Asturias, including two coastal locations, two central, and six locations along the Cantabrian Mountains, within national and natural parks, crucial hotspots for biodiversity conservation in Europe (Appendix A). In two of the locations, four different sites were sampled, making a total of 16 analysed sites. The names of these sites, their Nature 2000 codes, and their geographic information (altitudes and coordinates) are presented in Appendix A.

The plant samples consisted of recently emerged croziers and were collected in early spring 2023. Once cut, the plants were kept at 4 °C until they underwent lyophilization and were ground into powder before being stored at −20 °C.

### 4.2. Metabolite Analysis: Determination of Pterosins in Croziers from Bracken

The determination of pterosins B, A, and G was performed as described by [56], with several modifications. Briefly, for each sample, 40 mg of sample powder was mixed with 40 mL of a 20% methanol (MeOH) and 0.1 M ammonium acetate solution at pH 6. This mixture was agitated by vertical rotation in a Heidolph REAX 2 shaker at 74 rpm and 4 °C for 20 min; it was then centrifuged at 3000× *g* for 5 min at 4 °C, and the supernatant was collected. Then, 20 µL of a stock solution (2500 µg/L) of Loganin (CAS Nº: 18524-94-2, from TCI Europe N.V., Zwijndrecht, Belgium) was added as an internal standard to 480 µL of the supernatant [93]. This mixture was filtered with a 0.22 µm and 13 mm diameter polyvinylidene difluoride filter (PVDF). Chemical compounds PtrA (CAS Nº 35910-16-8) and PtrB (CAS Nº 34175-96-7) obtained from ChemFaces (Wuhan, China) were used as standards. All these compounds were separated and quantified by ultra-high-performance liquid chromatography coupled with tandem mass spectrometry detectors (UHPLC-MS/MS) in a 6460 Triple Quad LC/MS (Agilent Technologies) using Masshunter Workstation software (https://www.agilent.com/en/promotions/masshunter-mass-spec, accessed on 22 July 2025, Agilent Technologies, Madrid, Spain). The ultra-high-performance liquid chromatography (UHPLC) was carried out with the generated samples, using 0.1% formic acid in high-purity MilliQ water (*v*/*v*) as mobile phase A and 0.1% formic acid in acetonitrile (Romil UpS—Ultra Purity Solvent) as mobile phase B. An Agilent InfinityLab Poroshell 120 EC-C18 column (2.7 µm particle size, 3 × 50 mm i.d.), paired with a precolumn of identical specifications, was used for the stationary phase. The peristaltic pump was set at a constant flow rate of 1 mL/min, and the gradient was set as follows: 0–1 min 10% B; 1–3 min 35% B; 4–4.5 min 95% B; 4.6–5 min 10% B. A volume of 20 µL was injected, and the column temperature was kept at 35 °C. For the ionization and quantification stages, the Agilent 6460 was used in Dynamic Multiple Reaction Monitoring mode (dynamic MRM), after positive electrospray ionization (ESI), and 3200 V. The ionization parameters used were 13 L/min as the drying gas flow rate and 40 psi as the nebulizer pressure. Mass transitions for the Dynamic MRM method are presented in Appendix A, whereas information about precursor ions, product ions, and collision energies used for each chemical is presented in Appendix A. Three independently prepared extracts from each sample were analysed, with three replicates per independent extract. A blank was measured after each sample to confirm the absence of cross-contamination.

### 4.3. Genotoxicity Analysis: SMART Assay

The eye *w*/*w* + SMART assay was performed with the Oregon K strains yellow and white (Ok-y and Ok-w, respectively) of *D. melanogaster* and surface treatments. Briefly, 60–70 ± 12 h old larvae descended from mass crosses between Ok-y females and Ok-w males, developed in Formula 4–24 Caroline Instant Drosophila Medium, were treated with 1.5 mL/bottle of different extract concentrations, and the hatched adult eyes were scored as described [59]. In every experiment, distilled water was used as the negative control, and 2.5 mM methylmethane sulfonate (MMS) was used as the positive control. Three hundred eyes were scored per sex and analysed for concentration and/or condition. The number of hatched flies per bottle was used as a semiquantitative toxicity parameter, and the frequency of eyes with at least one mutant spot (mosaic eyes) was used to determine genotoxic activity.

To treat the larvae, aqueous plant extracts were prepared before each experiment, mixing sample powder with sterile MilliQ water in sterile vials at concentrations of 1, 10, and 25 mg/mL. The vials were agitated by vertical rotation in a Heidolph REAX2 at maximum speed for 2.5 h at room temperature. At least two independent experiments were carried out for each sample.

The pH and viscosity (using a U-tube viscosimeter from Rheotek, Essex, UK) of the 10 mg/mL concentrations from all the samples were determined immediately after this time.

### 4.4. Surveys Obtained from Farmers

To assess the awareness of farmers regarding the impact of bracken on cattle, a comprehensive survey was conducted in Asturias among farmers assigned to municipalities within 9 of the 10 sampled locations (no farmers were identified in Location 6) (Appendix A). Surveys were conducted among local farmers with extensive farms. Extensive farming means that during the grazing season, from spring to autumn, farmers allow their livestock to graze freely in the mountain passes. In contrast, in the winter months, the animals are moved to a stable controlled environment. A total of 212 surveys were completed, gathered mainly at cattle fairs. The survey consisted of 14 questions aimed at gauging farmers’ knowledge about bracken and its risks. These questions requested information about the type of animals and their numbers and, more importantly, about whether the farmers (i) were aware of the damage that the bracken might cause, (ii) have seen cattle eating bracken and at what time of the year (spring, summer, autumn, or winter), (iii) have observed/detected any symptoms that could presumably be related to bracken poisoning (bleeding, cancer, blindness, or avitaminosis), (iv) have suffered the death of animals due to poisoning, and finally, (v) have obtained veterinary death certificates due to bracken.

### 4.5. Implementing a Tool to Map Bracken

A preliminary proof-of-concept experiment using remote sensing data was performed in this project. Field data acquisition was conducted in an area of 320 ha in the Montes de Urbiés. In this area, four vegetation types were defined: forest, bracken, grasses, and others, which included bare soil, roads, human uses, etc. A total of 54 sample points were included as ground-truth data.

The multispectral data were captured using a fixed-wing UAV (eBee, SenseFly, Saint-Sulpice, Switzerland) equipped with a MicaSense Rededge-MX 5-channel multispectral camera (Micasense, Inc., Seattle, WA, USA). Considering the phenology of bracken, the acquisition flights were performed in November 2023 and May 2024. A multispectral reference board was photographed, before flight, for radiation correction. The UAV employed RTK technology, so geometric correction was not necessary. Multispectral data were processed using Pix4D software (https://www.pix4d.com/software-guide/, accessed on 22 July 2025), obtaining reflectance bands. Different vegetation indices [94] were calculated using the obtained reflectance bands. Moreover, free LiDAR data for Spain, provided by the Spanish National Aerial Photography Programme (Plan Nacional de Ortofotografía Aérea, PNOA), with a density of 1 point/m^2^, were used to obtain vegetation height (CHM, crown height model). Finally, textural indices and differences between time-lapse indices were included as predictor variables. These training data were used to perform a machine learning classification, using random forest (RF). Test data were generated through the photointerpretation of 250 random points using the orthophotos of the flights. The developed process was included in an open-source tool to replicate in new areas and obtain distribution maps of bracken. The tool includes a post-processing step after the classification, reabsorbing surfaces less than 5 square metres and reclassifying pixels following a condition (e.g., pixels with height greater than 4 m were reclassified as forest, or areas classified as grasses but with a height greater than 0.3 m were reclassified as bracken). The final output is a vector file with the classifications of vegetation types.

### 4.6. Statistical Analyses

The data presented in this work are arithmetic means with their standard deviations (SDs) or standard errors (SEs), and also frequencies with their SDs, estimated as *p*·*q*/√N, where *p* is the estimated frequency of mosaic eyes, *q* is 1 − *p*, and N is the number of scored eyes [61].

Results of metabolite levels among samples were analysed with one-way ANOVA, with Student–Newman–Keuls (SNK) post hoc tests, to check differences among samples. Analysis of the frequencies of mosaic eyes were carried out with Chi-square tests. The results of hatched flies/bottle in the treatments were compared to those of the corresponding negative controls with Student’s *t* tests. Relationships between different parameters were performed with bivariate correlations and linear regression analyses. Linear regressions were performed after checking their good adjustment. All these statistical analyses were carried out with the IBM SPSS program (version 21.0.0.0). Regarding the analysis of data collected from the surveys, when data did not meet the assumptions of normality, Kruskal–Wallis tests followed by Dunn’s post hoc tests were performed.

## 5. Conclusions

The determination of pterosins A and B demonstrates the presence of CAU and PTA, respectively, and their corresponding dienones, in bracken plants from Asturias. Furthermore, aqueous extracts of these plants induced mutation and recombination events in somatic cells of *D. melanogaster larvae* in vivo, depending on the pterosin levels. Moreover, the geographical distribution of bracken in Asturias is linked to its danger, because both the pterosins levels and the genotoxic activity of different plants were related to the altitude and latitude at which the plants were collected. These facts confirm the risk posed by bracken, especially when it is eaten by animals, because of the problem that their death represents to the farming economy. Nevertheless, not all the surveyed farmers were aware of these risks, depending on their location. An RF model that, although not bad, might be improved was developed to identify some vegetation types, including bracken. This work demonstrates the necessity of making public the bracken risks in Asturias, requesting the intervention of local and regional authorities, and promoting the dissemination of best practices among farmers and the broader community. This commitment is essential to address the serious risk that has been identified.

## Figures and Tables

**Figure 1 ijms-26-07157-f001:**
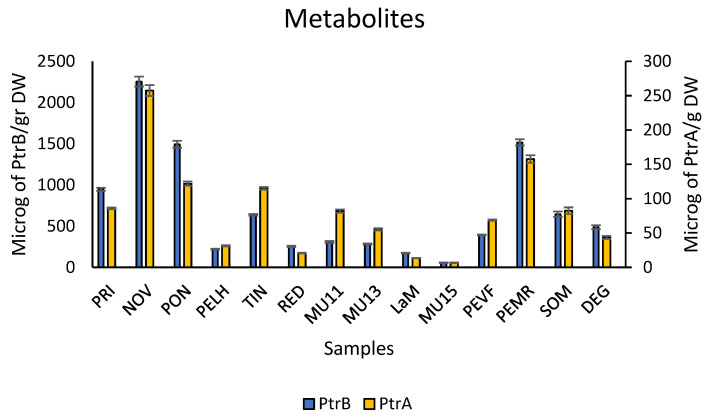
Levels of *Pteridium aquilinum* metabolites PtrA and PtrB in extracts from 14 plant samples. Data are arithmetic means of at least 3 independent extracts with their standard errors.

**Figure 2 ijms-26-07157-f002:**
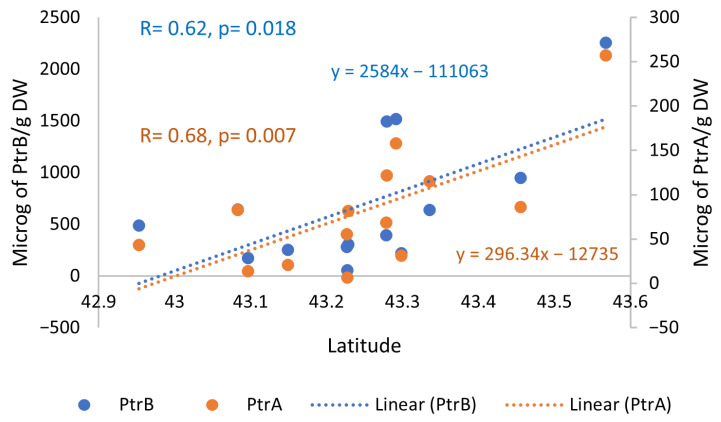
Relationships between metabolite levels and latitude. Positive statistically significant regression slopes and correlations for PtrB and PtrA. Regression equations are presented for all the metabolites.

**Figure 3 ijms-26-07157-f003:**
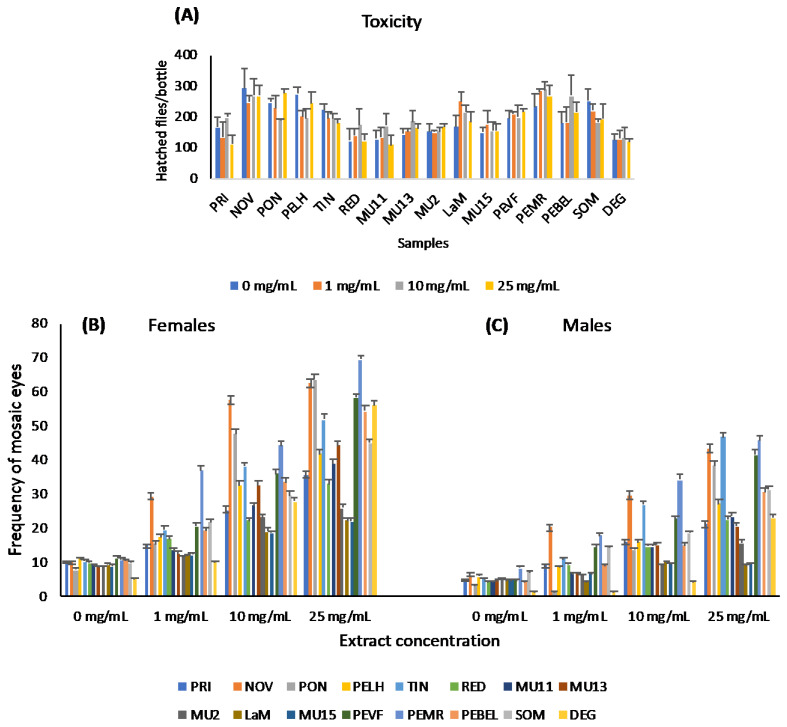
In vivo analysis of genotoxicity, with the SMART assay, of aqueous extracts from sixteen samples of plants. (**A**) Number of hatched flies per bottle as a measure of toxicity; values are arithmetic means of at least 4 bottles and their standard errors. (**B**,**C**) Frequencies of mosaic eyes, and their standard deviations, in females (**B**) and males (**C**).

**Figure 4 ijms-26-07157-f004:**
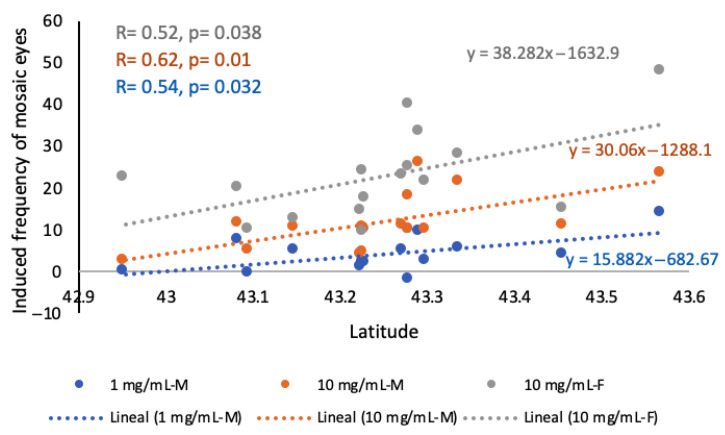
Relationships between induced frequencies of mosaic eyes and latitude in females and males. Positive statistically significant regression slopes and correlations (R, with d.f. = 14) for 1 and 10 mg/mL concentrations in males (in blue and orange, respectively), and for 10 mg/mL concentration in females (in grey). Regression equations are presented for all the concentrations.

**Figure 5 ijms-26-07157-f005:**
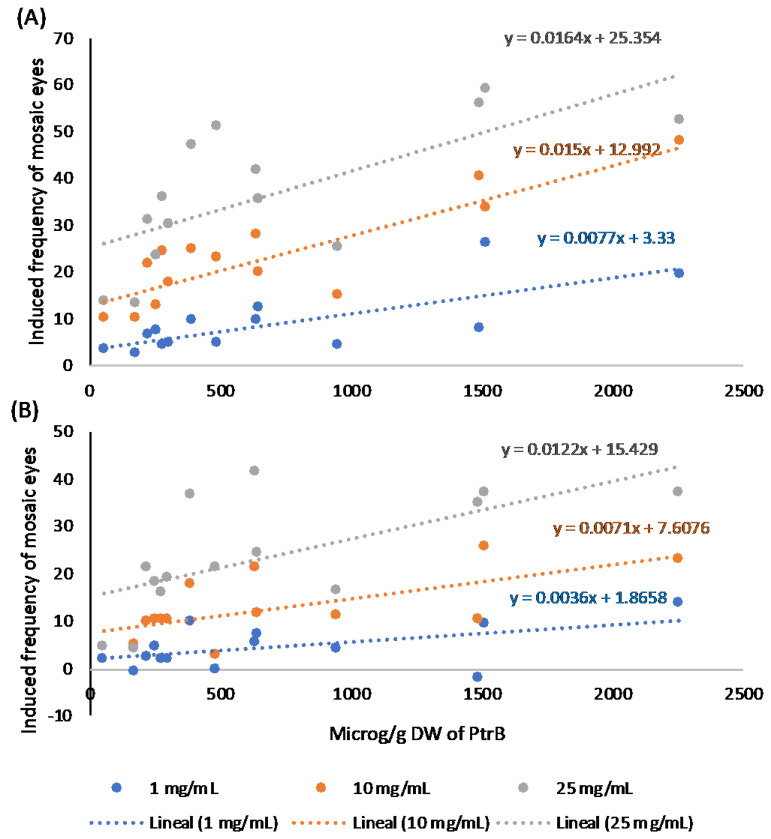
Relationships between induced frequencies of mosaic eyes and PtrB levels in females and males. (**A**) Statistically significant regression lines for the three analysed extract concentrations in females. (**B**) The same for 10 and 25 mg/mL extract concentrations in males. Regression equations are presented for the 1, 10, and 25 mg/mL concentrations in blue, orange and grey, respectively, in both sexes.

**Figure 6 ijms-26-07157-f006:**
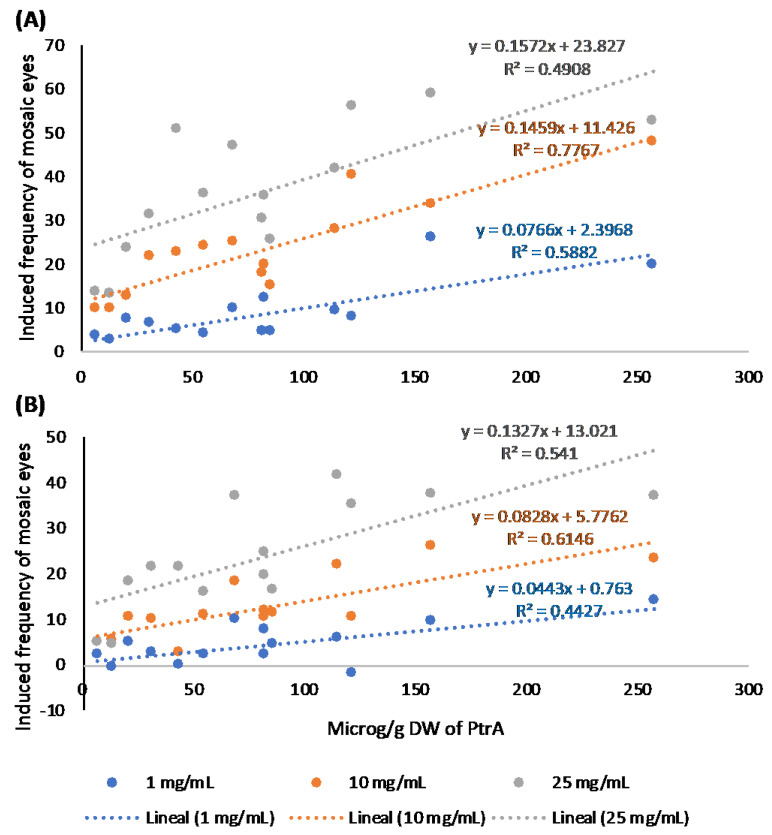
Relationships between induced frequencies of mosaic eyes and PtrA levels in females and males. (**A**) Statistically significant regression lines for the three analysed extract concentrations in females. (**B**) The same in males. Regression equations are presented for the 1, 10, and 25 mg/mL concentrations in blue, orange, and grey, respectively, in both sexes.

**Figure 7 ijms-26-07157-f007:**
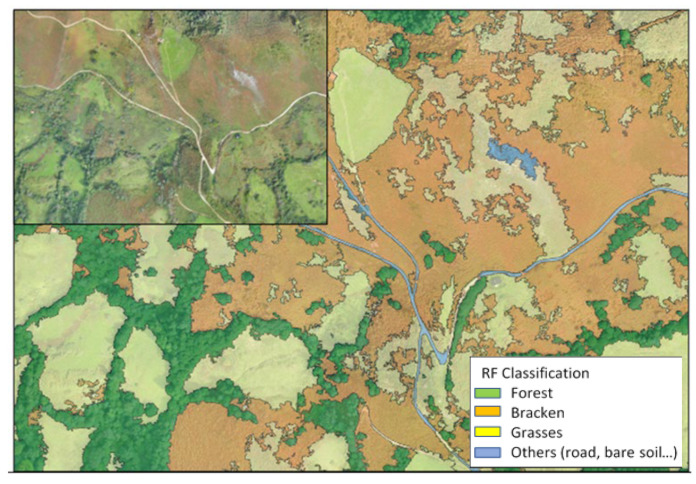
Distribution of vegetation types in the study area; the bracken distribution is coloured in orange (upper left: RGB original image).

**Table 1 ijms-26-07157-t001:** Analytical performance characteristics of the UHPLC-MS/MS used to detect pterosins A and B in extracts of bracken plants. LOD—limit of detection. LOQ—limit of quantification.

Chemical	μgL^−1^ Min	μgL^−1^ Max	R^2^	LOD (μgL^−1^)	LOQ (μgL^−1^)
PtrB	5	160	0.998	2.5	5
PtrA	0.75	80	0.994	0.046	0.37

## Data Availability

The original contributions presented in this study are included in the article/Appendix A. Further inquiries can be directed to the corresponding author.

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
