# Peer review of "Toxic Threats from the Fern Pteridium aquilinum: A Multidisciplinary Case Study in Northern Spain"

_ijms, 2025, doi:10.3390/ijms26157157_

Round 1
Reviewer 1 Report
Comments and Suggestions for Authors
In this work, the authors pursued a series of tasks to evaluate the potential threats bracken poses to the environment as well livestock feeding on them. The authors quantified pterosins A and B toxins in young tissue of bracken using analytical techniques, analyzed genotoxicity of bracken extracts, performed a large-scale survey to assess the farmers awareness and mapped spatial distribution of bracken in Northern Spain. The authors were successful in achieving these goals and presenting their data. I find the work interesting and novel though the authors had previously published a similar work but in a much-limited capacity.
I have a few comments as follows:
Line 153: I don’t see any standard error (SE) for MR, LH, VF, MU11, MU13 and MU15 bars in Figure 1.
Lines 201-203: The authors stated that the samples in Figure 3C were significantly higher than the corresponding negative controls at 10mg/mL concentration except for DEG. It seems to me that DEG is statistically greater than the corresponding negative control.
I would recommend the authors to cite their previous publication on the same matter as shown below:
Fernández, H., Sierra, L.M. (2022). Pteridium aquilinum: A Threat to Biodiversity and Human and Animal Health. In: Marimuthu, J., Fernández, H., Kumar, A., Thangaiah, S. (eds) Ferns. Springer, Singapore.
https://doi.org/10.1007/978-981-16-6170-9_30
Author Response
ANSWERS TO REVIEWERS
Many thanks for your effort in helping us to improve the manuscript. We have very much appreciated your comments. Thanks indeed.
-Line 153: I don’t see any standard error (SE) for MR, LH, VF, MU11, MU13 and MU15 bars in Figure 1.
- The Reviewer was right, there was a problem with this figure. Now, we have changed it to a new version with all the data
-Lines 201-203: The authors stated that the samples in Figure 3C were significantly higher than the corresponding negative controls at 10mg/mL concentration, except for DEG. It seems to me that DEG is statistically greater than the corresponding negative control.
- This Figure 3C was also changed because there was a mistake with the colours of the last three samples. In addition, as an answer to the question, the value of the chi-square test for 10 mg/mL concentrations in males, in the DEG sample, was 3.02 with correction for continuity, and its p-value was 0.0825. We have included this information in the text, in brackets.
-I would recommend that the authors cite their previous publication on the same matter as shown below:
Fernández, H., Sierra, L.M. (2022). Pteridium aquilinum: A Threat to Biodiversity and Human and Animal Health. In: Marimuthu, J., Fernández, H., Kumar, A., Thangaiah, S. (eds) Ferns. Springer, Singapore. https://doi.org/10.1007/978-981-16-6170-9_30.
- We very much appreciate the suggestion. We have included this work as reference number 14.
Reviewer 2 Report
Comments and Suggestions for Authors
The manuscript addresses the ecotoxicological properties of Pteridium aquilinum and the risks this poses to farmers' cattle. This study is highly important, given the widespread nature of this invasive fern and its known carcinogenic compounds. This multidisciplinary investigation provides valuable insights into the toxicological landscape of Pteridium aquilinum. The authors have combined modern methods of chemical analysis and genotoxic bioassays with socio-economic surveys and UAV-based remote sensing to create an interesting, multidisciplinary study.
Nevertheless, the work invites a number of questions and comments:
- How does the accumulation of pterosins change throughout the year? What are the levels like in the croziers in the early spring compared to other seasons? Is this the point at which the content is at its maximum?
- Is anything known about the geographical genetic variability of bracken? Could this be a factor in determining differences in toxin accumulation?3.
- Probably need to discuss the possible accumulation of bracken toxins in animal tissues.
- The author should indicate why linear models were chosen. It may be necessary to specify p and confidence intervals for the coefficients of regression models.
- Not all abbreviations are deciphered (e.g., LOD, LOQ in Table 1).
- It is worth putting the values of R and p on all figures.
The paper will be ready for publication after a minor revision.
Author Response
Many thanks for the effort you made, helping us to improve the manuscript. We have very much appreciated all the comments.
-How does the accumulation of pterosins change throughout the year? What are the levels like in the croziers in the early spring compared to other seasons? Is this the point at which the content is at its maximum?
1. We do not yet have information about metabolite levels, but we have information about the genotoxic activities of extracts from plants collected in autumn: they are statistically lower than those of extracts from plants collected in April at the same sites. We plan to include this information, together with other data not yet analyzed, in a future manuscript.
- Is anything known about the geographical genetic variability of bracken? Could this be a factor in determining differences in toxin accumulation?
2. We do not know the answer to the first question, although we have not found information about it. However, the answer to the second question should be YES, genetic variability should be an important factor in determining differences in metabolite production and accumulation. This type of study should be the next step in the analysis of Pteridium aquilinum.
-Probably need to discuss the possible accumulation of bracken toxins in animal tissues.
3. To our knowledge, there is no information about the accumulation of bracken metabolites in cow tissues, only in milk, and it is already mentioned in the manuscript.
- The author should indicate why linear models were chosen. It may be necessary to specify p and confidence intervals for the coefficients of regression models.
4. Since the dose-response relationships seem to be linear, linear regressions were carried out and, after checking their good adjustment, they were performed in all the analyses, confirming a very good fit in all the cases. The p-values were always indicated, either in the figures or in the text. We did not include confidence intervals because, having p-values, we thought they were not necessary. In addition, it will not be possible to include them on the graphs, as there is not enough space.
-Not all abbreviations are deciphered (e.g., LOD, LOQ in Table 1).
5. Now, the abbreviations are deciphered in Table 1.
It is worth putting the values of R and p on all figures.
6. In Figures 5 and 6, there is not enough free space to include these values. With this option, the figures would be too full, and it would be difficult to see the results. These data are included in the text of the manuscript.